# Role of matrix metalloproteinase-9 in neurodevelopmental deficits and experience-dependent plasticity in *Xenopus laevis*

Sayali V Gore[1], Eric J James[1], Lin-chien Huang[2], Jenn J Park[1], Andrea Berghella[1], Adrian C Thompson[1], Hollis T Cline[2], Carlos D Aizenman[1]*

[1]Department of Neuroscience, Brown University, Providence, United States; [2]The Scripps Research Institute, La Jolla, United States

**Abstract** Matrix metalloproteinase-9 (MMP-9) is a secreted endopeptidase targeting extracellular matrix proteins, creating permissive environments for neuronal development and plasticity. Developmental dysregulation of MMP-9 may also lead to neurodevelopmental disorders (ND). Here, we test the hypothesis that chronically elevated MMP-9 activity during early neurodevelopment is responsible for neural circuit hyperconnectivity observed in *Xenopus* tadpoles after early exposure to valproic acid (VPA), a known teratogen associated with ND in humans. In *Xenopus* tadpoles, VPA exposure results in excess local synaptic connectivity, disrupted social behavior and increased seizure susceptibility. We found that overexpressing MMP-9 in the brain copies effects of VPA on synaptic connectivity, and blocking MMP-9 activity pharmacologically or genetically reverses effects of VPA on physiology and behavior. We further show that during normal neurodevelopment MMP-9 levels are tightly regulated by neuronal activity and required for structural plasticity. These studies show a critical role for MMP-9 in both normal and abnormal development.

*For correspondence:
carlos_aizenman@brown.edu

Competing interests: The authors declare that no competing interests exist.

## Introduction

Autism spectrum disorder (ASD) encompasses a highly heterogeneous set of neurodevelopmental conditions characterized by a complex behavioral phenotype and deficits in both social and cognitive functions. Over 100 gene mutations have been associated with ASD suggesting very different underlying etiologies (*Betancur, 2011*; *Betancur and Coleman, 2013*; *Butler, 2018*; *Swanwick and Banerjee-Basu, 2011*; *Grove et al., 2019*). This complex neurodevelopmental disorder, which is influenced by both genetic and environmental factors, results from profound changes in brain function and connectivity. The pathophysiology of ASD is often attributed to abnormal synaptic maturation and plasticity, defects in microcircuitry organization, disruption of the brain excitation to inhibition balance, local overconnectivity and long-range underconnectivity throughout the brain (*Kumar et al., 2019*; *Markram and Markram, 2010*). Even though some aspects of the etiology of ASD have been studied, how ASD-related behaviors arise from synaptic dysfunction and abnormal neural circuit development remain elusive.

Studies in both humans (*Bromley et al., 2013*; *Christensen et al., 2013*) and animal models (*Markram et al., 2008*; *Rodier et al., 1996*; *Roullet et al., 2010*; *Schneider et al., 2006*; *Schneider and Przewłocki, 2005*) have shown that prenatal exposure to valproic acid (VPA), a common antiepileptic drug, results in a higher incidence of ASD in humans or ASD-like deficits in animals. In rodents, in utero exposure to VPA results in autistic-like behaviors in offspring; namely decreased socialization, increased repetitive behaviors, and hypersensitivity to sensory stimuli

(*Kataoka et al., 2013*; *Kim et al., 2011*; *Mehta et al., 2011*; *Moldrich et al., 2013*). These behavioral deficits are accompanied by corresponding deficits in brain physiology, including increased local recurrent connectivity, synaptic activity, and decreased intrinsic neuronal excitability (*Markram and Markram, 2010*). *Xenopus* tadpoles when exposed to VPA have abnormal sensory-motor and schooling behavior which is accompanied by hyper-connected neural networks in the optic tectum, increased excitatory and inhibitory synaptic drive, elevated levels of spontaneous synaptic activity and decreased neuronal intrinsic excitability (*James et al., 2015*). Consistent with these findings, VPA-treated tadpoles also have increased seizure susceptibility and decreased acoustic startle habituation (*James et al., 2015*). VPA, a known histone-deacetylase inhibitor, causes these effects by hyperacetylation of gene promoters, and thus, by altering gene expression of various targets (*Ghodke-Puranik et al., 2013*; *Terbach and Williams, 2009*). Preliminary studies in *Xenopus* using microarrays and qPCR identified MMP-9 upregulation as a potential downstream effect of VPA exposure, suggesting that dysregulation of this protein may be important for mediating effects of VPA.

MMP-9 is a secreted zinc-dependent extracellular endopeptidase that can cleave ECM and several cell surface receptors facilitating synaptic and circuit level reorganization (*Ethell and Ethell, 2007*; *Niedringhaus et al., 2012*; *Reinhard et al., 2015*). Early formation of proper synaptic connections and neural circuits is heavily regulated by the pericellular environment, and dysregulation of proteins important for extracellular matrix function could disrupt this process resulting in circuit abnormalities associated with ASD (*Choi, 2018*; *Ferrer-Ferrer and Dityatev, 2018*). During development, experience-dependent activity during critical periods can drive neural plasticity and MMP-9 plays an important role in this critical period plasticity (*Murase et al., 2017*; *Oliveira-Silva et al., 2007*; *Reinhard et al., 2015*; *Spolidoro et al., 2012*). Furthermore, MMP-9 is thought to play a role in adult neurogenesis and synaptic plasticity in the adult brain, and has been implicated in neural circuit formation (*Fujioka et al., 2012*; *Peixoto et al., 2012*; *Pielecka-Fortuna et al., 2015*; *Wang et al., 2008*).

Previous studies from the lab have utilized the *Xenopus laevis* tadpole CNS as a model for understanding the synaptic origin of neurodevelopmental disorders (*James et al., 2015*; *Pratt and Khakhalin, 2013*; *Truszkowski et al., 2016*). The optic tectum, a region in *Xenopus* CNS, receives converging inputs from multiple sensory modalities and is responsible for carrying out multisensory integration (*Deeg and Aizenman, 2011*; *Pratt and Aizenman, 2009*, *Pratt and Aizenman, 2007*). The retinotectal circuit and local circuits within the optic tectum are known to refine during development in an activity-dependent manner and mediate robust visually-guided behaviors that are very sensitive to abnormal circuit development (*Dong et al., 2009*; *Lee et al., 2010*), providing a strong link between neural circuit dysfunction and behavior and provide a robust model for understanding neurodevelopmental disorders at various levels of organization - from synapses to behavior. The present work investigates (1) whether upregulation of MMP-9 by whole-brain electroporation mimics electrophysiological effects induced by VPA, (2) whether downregulation of MMP-9 using a pharmacological agent or an antisense morpholino directed against MMP-9 rescues VPA-induced electrophysiological and behavioral effects, and (3) whether MMP-9 is required for activity-dependent structural plasticity.

## Results

### MMP-9 overexpression increases spontaneous synaptic activity and network connectivity, an effect also observed after VPA exposure

Exposure to VPA increases spontaneous synaptic activity, network connectivity and excitability in the developing tectum in *Xenopus* (*James et al., 2015*). Previous microarray studies from the lab have shown that VPA exposure results in striking alterations in gene expression, including a greater than two-fold increase in matrix metalloproteinase 9 (MMP-9). Here, we validated this finding by performing qPCR on tectal samples of VPA-exposed tadpoles and found a several-fold increase in mRNA levels for MMP9 [increase in MMP9 mRNA: 2.7±0.3 fold increase over control, n=three replicates]. This suggested the hypothesis that chronically elevated levels of MMP-9 may mediate some of the effects of VPA by promoting excess plasticity leading to local hyperconnectivity. Thus, we first tested whether overexpression of MMP-9 mimics the VPA-induced effects on synaptic connectivity and

excitability in the developing tectum. MMP-9 was overexpressed in tectal neurons of stage 42 tadpoles (approximately 7 days post-fertilization) using whole-brain electroporation, which results in expression in a small percent of tectal neurons (*Haas et al., 2002*). We performed whole-cell patch-clamp recordings from MMP-9-expressing tectal neurons of stage 47–48 tadpoles (approximately 15 days post-fertilization) using an ex vivo whole-brain preparation. One-way ANOVA showed significant differences in sEPSC frequencies between groups [sEPSCs frequency (events/s): p = 0.002, sIPSCs frequency (events/s): p = 0.03]. Post-hoc pairwise comparison indicated a significant increase in the frequency of sEPSCs [*sEPSCs frequency (events/s): GFP control, 1.37 ± 0.39, n = 11; MMP-9 transfected, 6.74 ± 1.54, n = 15; MMP-9 non-transfected, 2.92 ± 0.63, n = 19; p (GFP control and MMP-9 transfected) = 0.004; p (MMP-9 non-transfected and MMP-9 transfected) = 0.02*] and sIPSCs [*sIPSCs frequency (events/s): GFP control, 5.22 ± 0.97, n = 11; MMP-9 transfected, 8.09 ± 0.73, n = 13; MMP-9 non-transfected, 6.34 ± 0.52, n = 15; p (GFP control and MMP-9 transfected) = 0.03; p (MMP-9 non-transfected and MMP-9 transfected) = 0.25*] in the transfected MMP-9 overexpressing tectal cells compared with GFP controls and non-transfected neighbors (*Figure 1A,B*). There were no significant differences in the sEPSC amplitude [*sEPSCs amplitude (pA): GFP control, 5.53 ± 0.56, n = 11; MMP-9 transfected, 6.18 ± 0.33, n = 15; MMP-9 non-transfected, 6.51 ± 0.99, n = 19; p = 0.45*] or sIPSCs amplitude [*sIPSCs amplitude (pA): GFP control, 5.22 ± 0.33, n = 11; MMP-9 transfected, 5.99 ± 0.46, n = 13; MMP-9 non-transfected, 4.93 ± 0.43, n = 15; p = 0.19*] (*Figure 1C*), consistent with VPA-induced effects (*James et al., 2015*). Increased synaptic transmission observed in MMP-9 overexpressing tectal neurons is consistent with dysfunctional synaptic plasticity of sensory and local synaptic connections during development. Various theories of ASD and other neurodevelopmental disorders suggest that hyperconnected and hyperexcitable networks are the hallmarks of these disorders (*Supekar et al., 2013*; *Takarae and Sweeney, 2017*). In the developing *Xenopus* tectum, spontaneous barrages of recurrent synaptic activity are often used as a measure of intra-tectal connectivity and network excitability (*James et al., 2015*; *Pratt and Aizenman, 2007*). To gather more information about the tectal connectivity and excitability, we counted the number of spontaneous recurrent barrages occurring in MMP-9 overexpressing tectal cells as well as from GFP control cells. Increased number of spontaneous recurrent barrages were observed in both transfected and non-transfected cells from MMP-9 overexpressing animals, compared to GFP controls, indicating persistent local connectivity which usually is pruned during development (*Pratt et al., 2008*) [*Mean number of barrages/min: GFP control, 0.1 ± 0.1, n = 11; MMP-9 transfected, 1.13 ± 0.37, n = 15; MMP-9 non-transfected, 1.10 ± 0.40, n = 19*] (*Figure 1D*). The increased amount of barrage activity in non-transfected neurons, indicates that a small number of MMP-9-overexpressing cells are also sufficient to drive enhanced activity in the whole tectal network.

## Pharmacological inhibition of MMP-9 rescues VPA-induced effects

In order to confirm whether MMP-9 upregulation mediates the neurodevelopmental effects of VPA exposure, we tested whether downregulation of MMP-9 by introduction of a pharmacological inhibitor of MMP-9 (3 μM SB-3CT, *Brown et al., 2000*) would rescue VPA-induced effects. The sEPSC frequency was significantly different across different treatments [*sEPSC frequency (events/s): p = 0.02*]. Consistent with prior observations, we observed a significant increase in the frequency of sEPSCs in the VPA group compared to the control. Addition of SB-3CT to VPA resulted in a reversal of the VPA-induced increase in sEPSC frequency, while SB-3CT alone did not show any significant effect of sEPSC frequency [*sEPSC frequency (events/s): control, 3.13 ± 0.48, n = 14; VPA, 6.78 ± 1, n = 21; VPA + SB-3CT, 4.3 ± 0.67, n = 22; SC-3CT, 4.46 ± 1.19, n = 11, P (control and VPA) = 0.004; P(VPA and VPA+SB-3CT) = 0.02*] (*Figure 2A,B*). There were no significant differences in sEPSC amplitude [*sEPSC amplitude (pA): control, 5.27 ± 0.36, n = 14; VPA, 5.83 ± 0.35, n = 21; VPA + SB-3CT, 5.94 ± 0.22, n = 22; SC-3CT, 5.11 ± 0.41, n = 11; p = 0.24*] (*Figure 2C*). We further assessed changes in local excitatory connectivity by measuring evoked synaptic recurrent activity by electrically stimulating the optic chiasm using a bipolar stimulating electrode, and measuring the total charge during a time window which consists of predominantly recurrent activity (*Pratt et al., 2008*). One-way ANOVA showed significant differences across treatments for evoked responses [*Total charge (pA. sec): p = 0.03*]. We observed that maximal excitatory responses were elevated in VPA-treated tadpoles compared to the control group; and this effect was reversed in the VPA + SB-3CT group [*Total charge (pA.sec): control, 1160.61 ± 521.17, n = 8; VPA, 4163.43 ± 574.77, n = 16; VPA + SB-3CT, 1780.89 ± 479.29, n = 5; SC-3CT, 2655.15 ± 718.41, n = 13; p (control and VPA) = 0.01; p (VPA and*

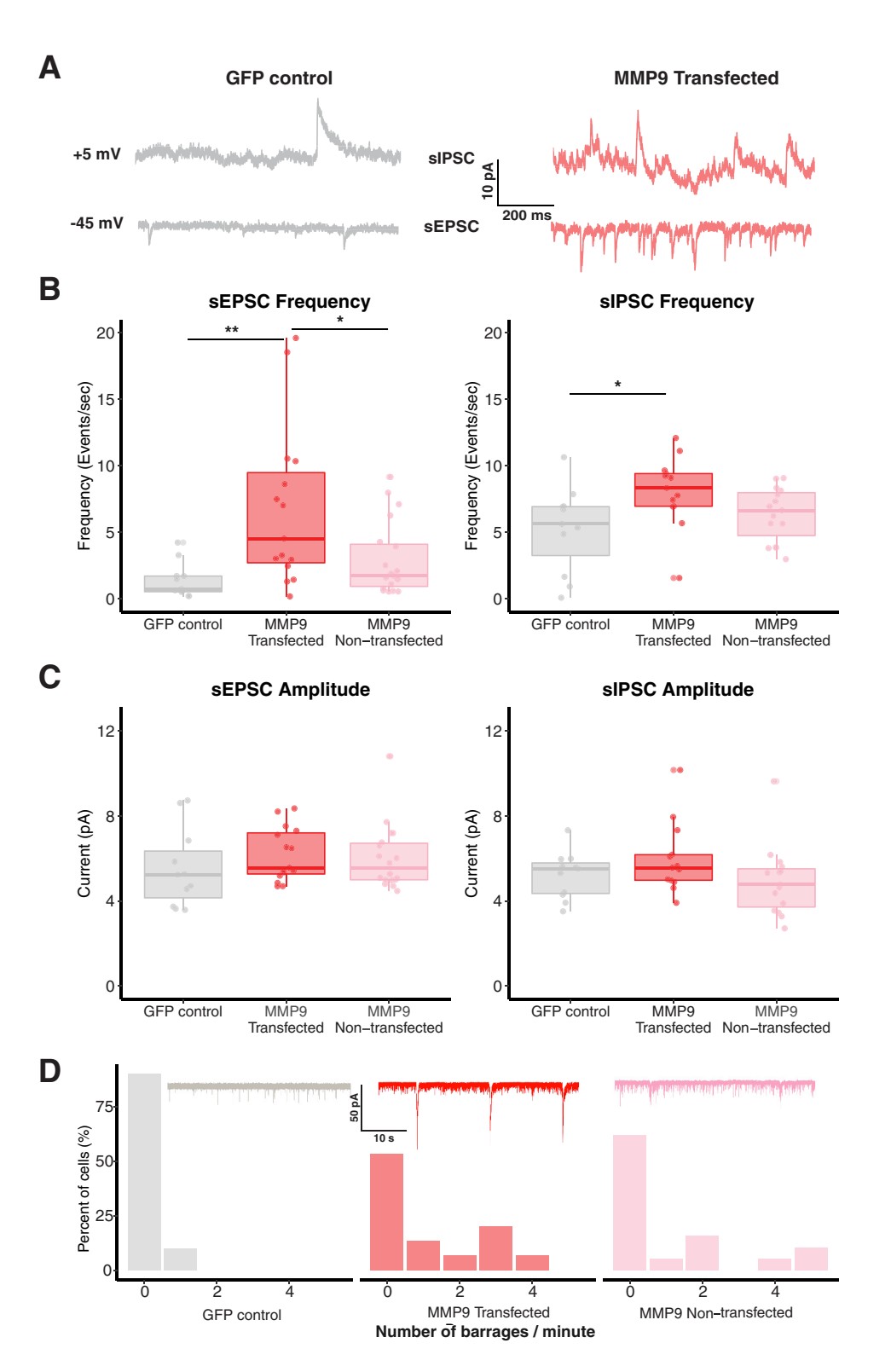

**Figure 1.** MMP-9 overexpression increases spontaneous synaptic activity and network connectivity. (**A**) Representative sEPSCs and sIPSCs recorded at −45 mV and +5 mV, respectively, from control and MMP-9 transfected cells. (**B**) Boxplots showing frequency of sEPSCs and sIPSCs for GFP control, MMP-9 transfected and MMP-9 non-transfected groups. The sEPSC and sIPSC frequencies are significantly enhanced in the MMP-9 transfected cells in comparison to the GFP control. Each dot represents data from one cell, the box represents median and interquartile range (IQR) and whiskers

*Figure 1 continued on next page*

Figure 1 continued

represent the range of data. (**C**) Box plot showing sEPSC and sIPSC amplitudes for each group. The sEPSC and sIPSC amplitudes do not vary significantly between control, MMP-9 transfected and MMP-9 non-transfected groups. Data are represented as median ± IQR. (**D**) Histogram showing relative barrage frequency for each group. The inset shows representative traces of barrages from each experimental group. MMP-9 transfected and non-transfected groups showed a higher number of barrages (recorded at −45 mV) than the GFP control group. * p < 0.05, ** p < 0.005. The online version of this article includes the following source data for figure 1:

**Source data 1.** MMP9 overexpression data.

VPA+SB-3CT) = 0.03] (**Figure 2D**). Increased number of spontaneous recurrent barrages were observed in the VPA-reared tadpoles, compared to the GFP control; an effect rescued when VPA-rearing media was supplemented with SB-3CT [*Number of barrages: control, 0.5 ± 0.19, n = 14; VPA, 1.19 ± 0.31, n = 21; VPA+SB-3CT, 0.59 ± 0.19, n = 22; SB-3CT, 0.36 ± 0.13, n = 11*] (**Figure 2E**). These data suggest that pharmacological inhibition of MMP-9 reverses increases in excitatory transmission observed in VPA treated animals, but has no effects on untreated controls, suggesting that basal MMP-9 levels are low.

One important question is the nature of the increase in spontaneous synaptic transmission observed after VPA exposure. Changes in synaptic transmission are unlikely to be due to increased intrinsic excitability of tectal neurons, as this had been shown to decrease after VPA exposure (**James et al., 2015**), and thus would lead to less network excitability. One prediction from this is that increases in spontaneous transmission are due to an increased number of synapses in the tectum after chronic VPA exposure. We directly tested this hypothesis by measuring synapse density by co-expressing GFP-tagged PSD95 to label postsynaptic densities (**Sanchez et al., 2006**) together with red fluorescent marker mCherry to label cell morphology. Although PSD-95 overexpression alone is known to increase synapse maturation (**El-Husseini et al., 2000**), we found that VPA-treated tadpoles expressed a significant increase in synapse density compared to controls (**Figure 2—figure supplement 1**). Together with the electrophysiological data, this experiment indicates that sEPSC frequency is a good proxy for synapse density, and that increased network hyperexcitability observed after VPA exposure likely arises due to increased density of synapses on tectal neurons.

## Morpholino targeted against MMP-9 rescues VPA-induced effects

As an alternative manipulation to SB-3CT inhibition of MMP-9 activity, we genetically downregulated levels of MMP-9 using an antisense morpholino oligo (MO) targeted against MMP-9 to test whether we could reverse VPA-induced effects. Whole-brain electroporation of the anti-MMP-9 MO, resulted in a significant decrease in endogenous MMP-9 levels when compared to the scrambled control MO (*Relative MMP-9 levels for MMP-9 MO/ Control MO, 0.72 ± 0.048, n = 5, p = 0.004*) (**Figure 3A**). As an alternative measure for MO effectiveness, we overexpressed an MMP-9 construct containing a FLAG tag, and found that the MMP-9 MO could decrease overexpression levels (not shown). We observed a significant decrease in the frequency of sEPSCs in the VPA+MMP-9 MO group compared to the VPA+control MO group suggesting that addition of MMP-9 MO in the presence of VPA rescues VPA-induced effects (**Figure 3B**). In a separate set of experiments, electroporation of MMP-9 MO alone on untreated animals did not have any effect on sEPSC frequency when compared to control MO (**Figure 3C**). [*sEPSC frequency (events/s): VPA+control MO, 5.99 ± 0.79, n = 9; VPA+MMP-9 MO, 1.55 ± 0.28, n = 10; control MO, 6.09 ± 1.27, n = 11; MMP-9 MO, 6.55 ± 1.33, n = 18, p (VPA+control MO and VPA+MMP-9 MO) < 0.0001; p (control MO and MMP-9MO) = 0.81*]. There were no significant differences in sEPSC amplitude [*VPA+control MO, 6.75 ± 0.57, n = 9; VPA+MMP-9 MO, 6.3 ± 0.51, n = 10; control MO, 6.57 ± 0.52, n = 11; MMP-9 MO, 6.73 ± 0.61, n = 18, p (VPA+control MO and VPA+MMP-9 MO) = 0.61; p (control MO and MMP-9 MO) = 0.60*] (**Figure 3C**). We next examined changes in local excitatory connectivity by measuring evoked synaptic recurrent activity by electrically stimulating the optic chiasm. We observed that maximal excitatory responses were significantly reduced in the VPA+MMP-9 MO group in comparison to the VPA+control MO group [*Total charge (pA.sec): VPA+control MO, 14792 ± 2454.59, n = 9; VPA+MMP-9 MO, 2752.9 ± 390.4, n = 10, p (VPA+control MO and VPA+MMP-9 MO) < 0.0001*] (**Figure 3B**). The maximal excitatory responses were not significantly different for only control MO and MMP-9 MO groups [*Total charge (pA.sec): control MO, 2926.95 ± 994.03, n = 8; MMP-9 MO, 2295.82 ± 478.24, n = 9, p (control MO and MMP-9 MO) = 0.56*] (**Figure 3C**). The VPA+MMP-9 MO group showed

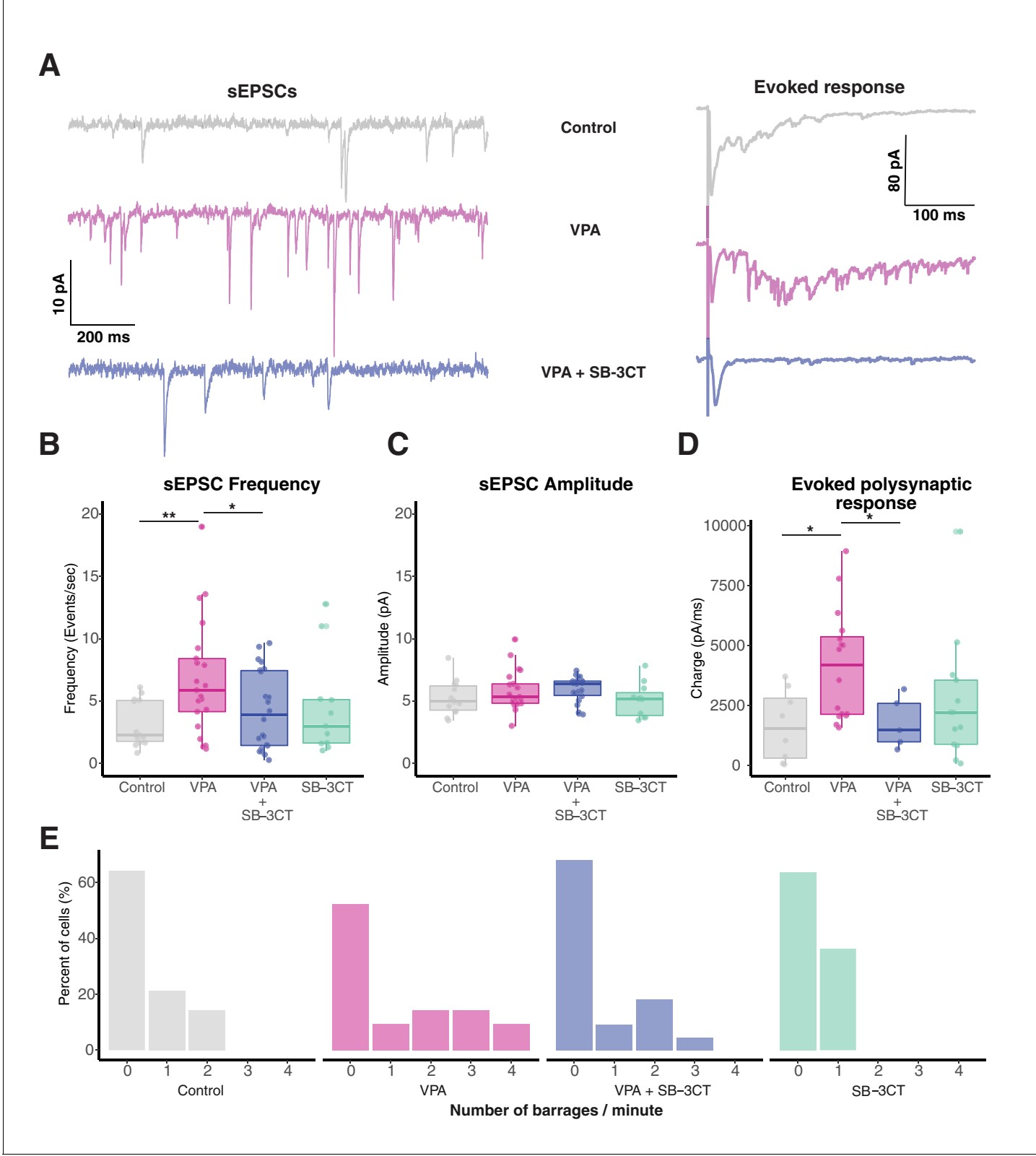

**Figure 2.** Pharmacological inhibition of MMP-9 rescues VPA-induced effects on synaptic transmission. (**A**) Representative sEPSCs (left) and evoked response (right) form control, VPA and VPA + SB-3CT-treated animals. (**B**) Boxplots showing frequency of sEPSCs for control, VPA, VPA + SB-3CT, and SB-3CT alone groups. Frequency of sEPSCs are significantly enhanced in the VPA group, an effect rescued by addition of SB-3CT. SB-3CT alone does not have any significant effect on sEPSC frequency. Data are represented as median ± IQR. (**C**) Boxplots showing sEPSCs amplitude for each group. *Figure 2 continued on next page*

*Figure 2 continued*

sEPSC amplitude does not change significantly between control and treatment animals. Data are represented as median ± IQR. (D) Boxplot showing total charge from evoked synaptic response over a 100 ms window for each group. Total charge from evoked synaptic response is significantly higher in VPA treated animals and this effect is rescued in the presence of SB-3CT. Data are represented as median ± IQR. (E) Histogram showing relative barrage frequency for each group. Higher number of barrages are observed for VPA treated animals than in any other group. * p < 0.05, ** p < 0.005. The online version of this article includes the following source data and figure supplement(s) for figure 2:

**Source data 1.** SB3CT source data.
**Figure supplement 1.** Exposure to VPA results in increased synapse density.
**Figure supplement 1—source data 1.** PSD95 overexression source data.

decreased frequency of barrages compared to the VPA +control MO group, suggesting that the downregulation of MMP-9 using MMP-9 MO rescues VPA-induced effects on spontaneous network activity [*Number of barrages: VPA+control MO, 1.88 ± 0.56, n = 9; VPA+MMP-9 MO, 0.7 ± 0.3, n = 10*]. The number of barrages were similar in only control MO and MMP-9 MO groups in the absence of VPA exposure [*control MO, 1.09 ± 0.31, n = 11; MMP-9 MO, 1.22 ± 0.31, n = 18*] (*Figure 3D*). Consistent with the SB-3CT findings, these data suggest that suppressing VPA-induced MMP-9 elevations, results in a reversal of VPA effects on excitatory synapse overconnectivity, but has little effect on baseline transmission in untreated animals.

## Inhibition of MMP-9 reverses VPA-induced behavioral effects

The results so far suggest that increasing levels of MMP-9 produces VPA-like hyperconnected excitable tectal networks while downregulation of MMP-9 rescues these effects. One consequence resulting from increased recurrent circuitry within the tectum, and possibly other brain regions, is that this circuitry can promote the generation of behaviors associated with hyperexcitability like increased seizure susceptibility or abnormal startle habituation response (*James et al., 2015*). We tested whether inhibition of MMP-9 could reverse these behavioral effects of VPA. To measure seizure susceptibility, tadpoles were exposed to 5 mM of the convulsant PTZ, over a period of 20 min, during which seizure activity was characterized. One-way ANOVA revealed significant differences for seizure frequency across different treatment groups [*seizure frequency (events/minute): p = 0.01; time to first seizure (s): p = 0.006*]. Consistent with prior work, VPA-reared tadpoles exhibited more frequent pharmacologically induced seizures compared to their matched controls, while animals reared in VPA+SB-3CT showed no increased seizure frequency, that is, a reversal of the VPA-induced effect [*seizure frequency (events/minute): control, 0.72 ± 0.04, n = 48; VPA, 0.90 ± 0.04, n = 45; VPA+SB-3CT, 0.8 ± 0.05, n = 34; p (control and VPA) = 0.004; p (VPA and VPA+SB-3CT) = 0.12*] (*Figure 4A*). This indicates that pharmacologically inhibiting MMP-9 during VPA exposure prevents VPA-induced changes in seizure susceptibility. As a second behavioral test, we measured acoustic startle habituation responses by presenting repeated acoustic stimuli over six bouts. Typically, tadpoles show habituation during each bout and across bouts (*James et al., 2015*). A repeated measure ANOVA revealed significant differences in startle speed across experimental groups [*p < 0.001], bouts [p < 0.001] and group*bouts [p < 0.001]* interaction. Post-hoc analysis showed significant differences between VPA vs. control and VPA vs VPA+SB3CT starting from bout 2 (Dunnett's). Control tadpoles showed decreased startle speed over repeated bouts, indicating habituation (*Figure 4B*). In VPA reared tadpoles, the animals showed reduced habituation. Co-exposure of SB-3CT and VPA reversed VPA-induced effects; although the startle habituation was significantly, but only partially restored relative to control levels. The sixth bout was measured with a 15 min delay between bouts, indicating long-term habituation, and this was also similarly affected across experimental groups.

## Enhanced visual activity increases MMP-9 levels and MMP-9 inhibition prevents visual-activity induced dendritic growth

Our data suggest that while inhibiting MMP-9 reverses the effect of VPA, this inhibition has no effect on control animals, suggesting that MMP-9 levels are low at baseline. One possibility is that during normal development, MMP-9 levels are tightly regulated by neural activity and are elevated transiently when there is a need for structural plasticity. Thus, periods of increased MMP-9 activity may correlate with experience-dependent synaptic reorganization during critical periods in the

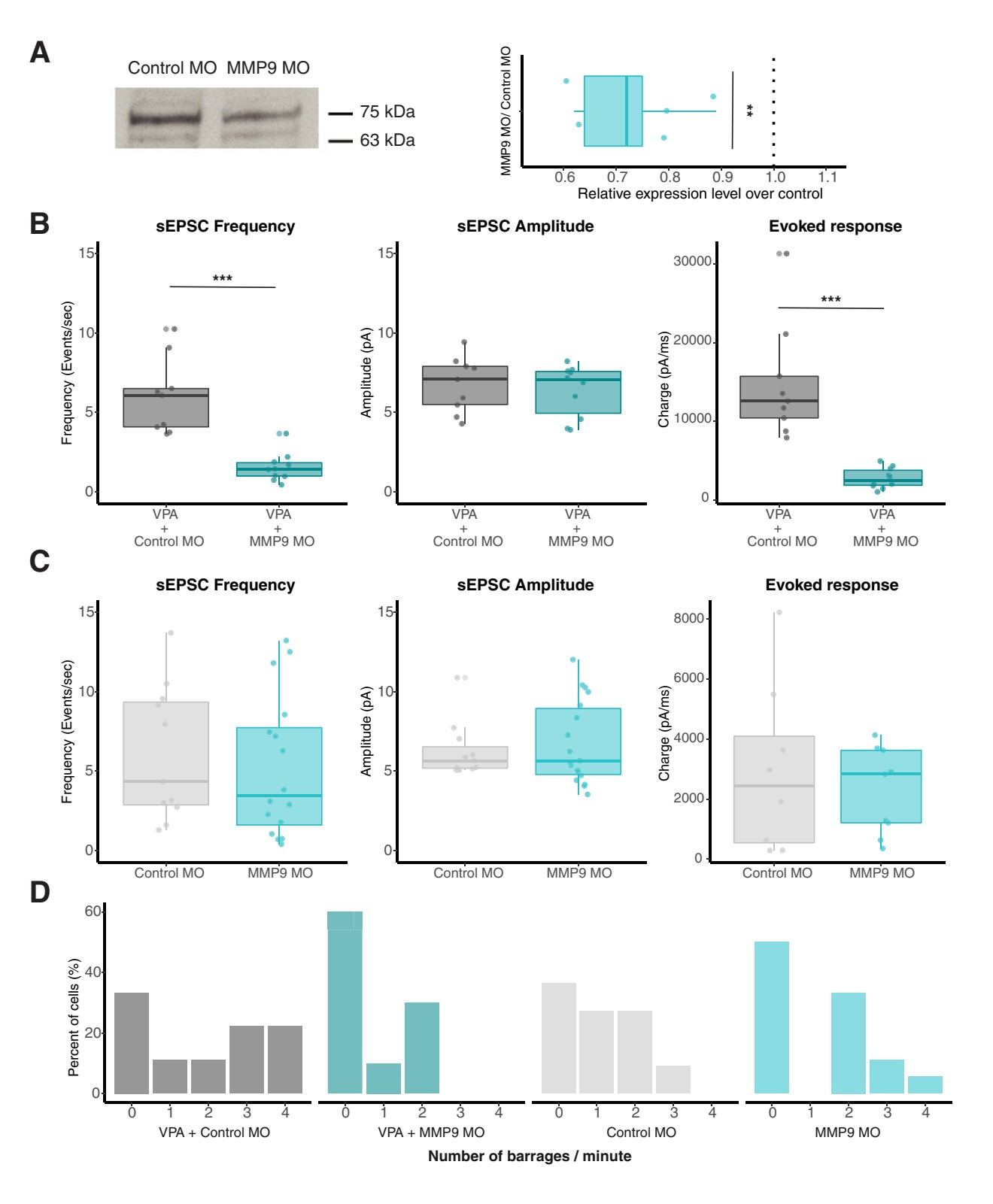

**Figure 3.** Morpholino targeted against MMP-9 rescues VPA-induced effects. (**A**) Western blot image (left) indicating MMP-9 levels (75kD band) in control (scrambled) MO and MMP-9 MO transfected groups. Boxplot (right) shows relative MMP-9 levels for control MO and MMP MO transfected tadpoles. MMP-9 MO group consistently shows relatively reduced MMP-9 levels compared to controls. The dotted line represents a ratio of 1 (no difference). The 63 kDa band is a non-specific band labeled by MMP-9 antibody that shows no change. (**B**) Boxplots showing sEPSC frequency,

*Figure 3 continued on next page*

*Figure 3 continued*

amplitude and evoked synaptic response from VPA + Control MO and VPA + MMP-9 MO groups. Frequency of sEPSCs and not amplitude are significantly enhanced in VPA + Control MO group. Total charge from evoked synaptic response is significantly higher in the VPA + Control MO compared to VPA + MMP-9 MO-treated animals. Data are represented as median ± IQR. (C) Boxplots showing sEPSC frequency, amplitude and evoked synaptic response from Control MO and MMP-9 MO alone. Control MO and MMP-9 MO alone does not differ significantly in sEPSC frequency or amplitude. MMP-9 MO alone does not show any significant effect on evoked response. Data are represented as median ± IQR. (D) Histogram showing relative barrage frequency for each group. Higher number of barrages are observed for VPA + control MO, and this effect is rescued by addition of MMP-9 MO. ** p < 0.005, *** p < 0.0001.

The online version of this article includes the following source data for figure 3:

**Source data 1.** Morpholino source data.

development of an organism (*Murase et al., 2017*). We used an assay for experience-dependent plasticity in the tectum, to test this hypothesis. Exposure of freely-swimming tadpoles to enhanced-visual stimulation for several hours is known to promote tectal neuron dendritic growth and synapto-genesis (*Aizenman and Cline, 2007*; *Sin et al., 2002*). We first tested whether this manipulation elevated MMP-9 levels in the brain. We exposed a group of tadpoles to enhanced visual activity in a custom-built chamber for 4 hr, another group was dark-exposed and a third group was left in normal light/dark rearing conditions, and then measured brain MMP-9 levels using a western blot. We found that exposure to enhanced visual activity resulted in significantly increased MMP-9 levels compared to dark and normal rearing conditions *[Ratio of MMP-9 levels: L/D, 1.34 ± 0.10, n = 5; L/N, 1.35 ± 0.10, n = 5; D/N, 1.04 ± 0.14, n = 5; p (L/D) = 0.03; p (L/N) = 0.03; p (D/N) = 0.79]* (*Figure 5A,B*).

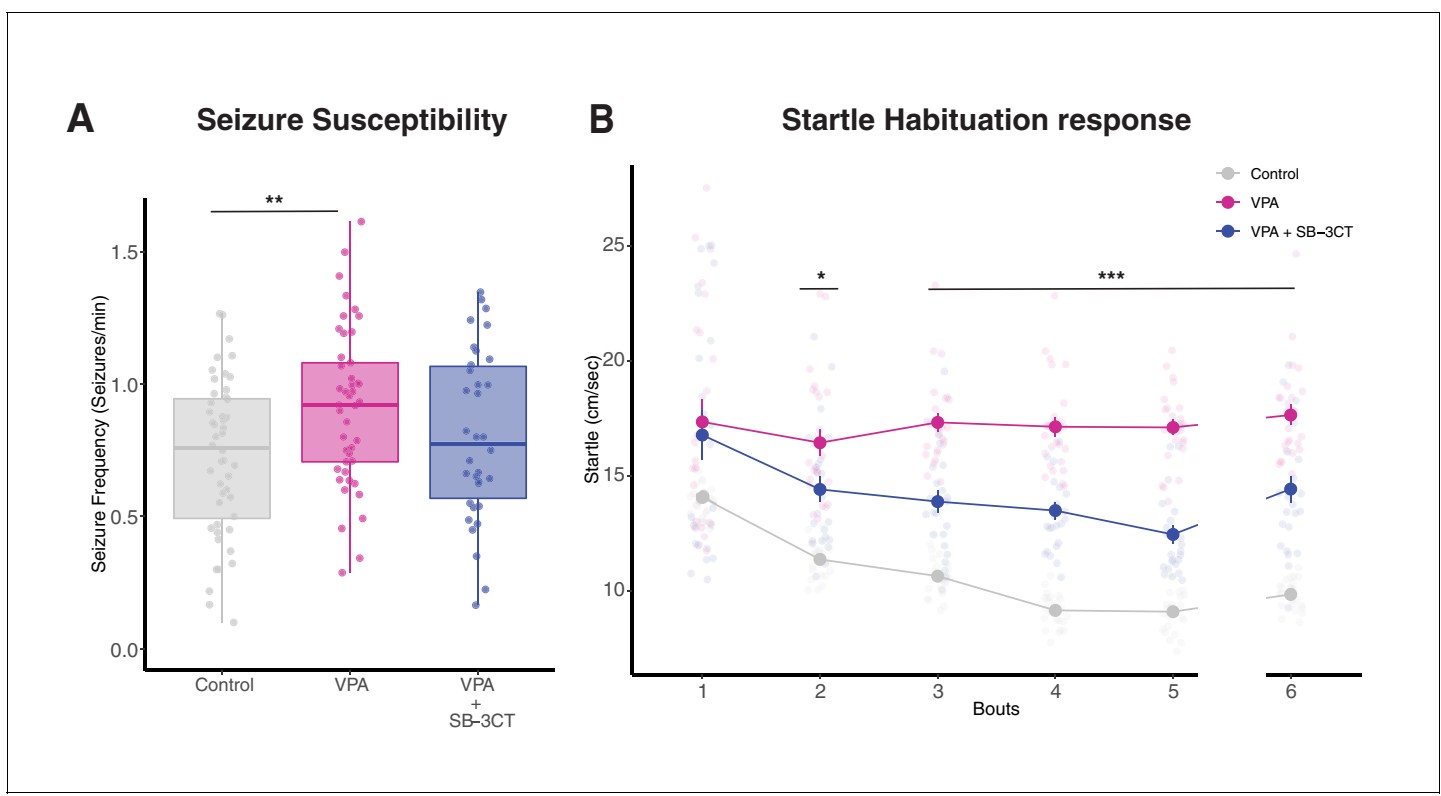

**Figure 4.** Inhibition of MMP-9 reverses VPA-induced behavioral effects. (A) Boxplots showing seizure frequencies for Control, VPA and VPA + SB-3CT groups. Seizure frequency is increased in the VPA group, but this effect is rescued by addition of SB-3CT. Data are represented as median ± IQR. (B) Line plot for startle response habituation for Control, VPA, and VPA + SB-3CT. VPA-treated animals show less habituation to startle response compared to control animals. Addition of SB-3CT to VPA reverses this effect. Bouts 1 to 5 are separated by 5 min time intervals, whereas bout six is separated from bout 5 by a 15-min time interval. Each dot represents average with standard error bars. * p < 0.05, ** p < 0.005, *** p<0.001.

The online version of this article includes the following source data for figure 4:

**Source data 1.** Behavior data.

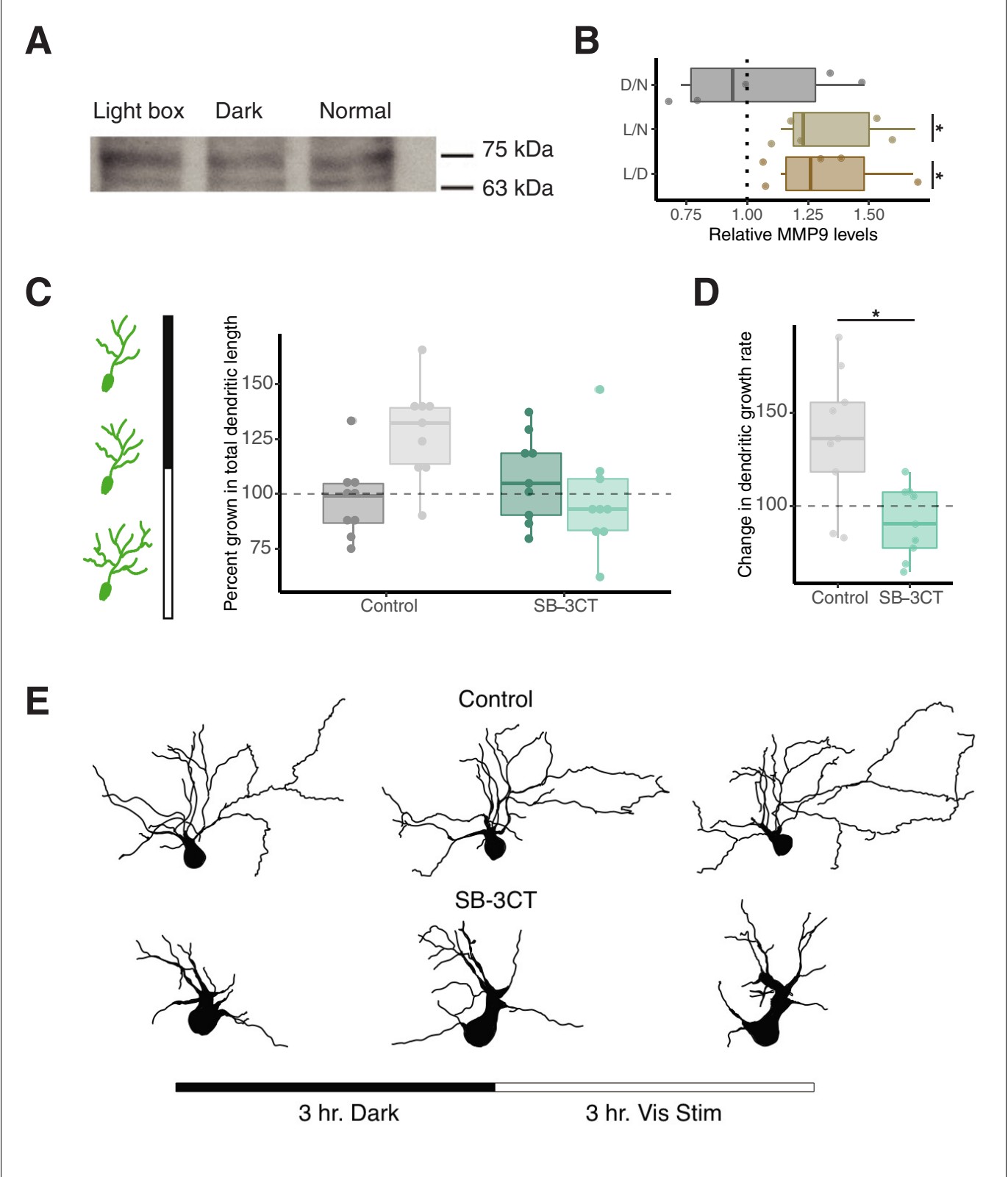

**Figure 5.** Enhanced visual activity increases MMP-9 levels and MMP-9 inhibition prevents visual-activity induced dendritic growth. (**A**) Western blot analysis specifying MMP-9 levels (75 kDa band) after lightbox, dark exposure and naive controls (normal). (**B**) Box plot showing relative MMP-9 levels for light box treatment (enhanced visual stimulation) vs dark exposed (L/D), light box vs naive controls (L/N) and dark exposed vs naive controls (D/N), based on western blot analysis. Enhanced visual activity using a light box experimental setup significantly increases relative MMP-9 levels compared to
*Figure 5 continued on next page*

**Figure 5 continued**

dark and normal rearing conditions. The dotted line represents a ratio of 1 (no difference). Data are represented as median ± IQR. (C) Left panel shows schematic of experimental design, three confocal imaging sessions were performed: one at the start of an experiment, a second after 3 hr of dark exposure and a third after 3 hr of enhanced visual stimulation. In control animals, visual stimulation enhances dendritic growth whereas SB-3CT prevents dendritic growth during this period. (D) SB-3CT significantly reduces the growth rate of dendrites during visual stimulation compared to the control group. Data are represented as median ± IQR. (E) Representative 2D tracings of 3D tectal cell reconstructions from control and SB-3CT groups are shown before any treatment (left), after 3 hr of dark exposure (middle) and after enhanced visual stimulation (right). * p < 0.05.

The online version of this article includes the following source data and figure supplement(s) for figure 5:

**Source data 1.** Visual stimulation source data.
**Figure supplement 1.** Exposure to VPA results in significantly less WFA staining intensity in comparison to control animals.
**Figure supplement 1—source data 1.** WFA staining data.

We then compared the effect of MMP-9 inhibition on visual-stimulation induced dendritic growth. We used electroporation to express GFP in single tectal neurons in vivo (*Bestman et al., 2006*), and then imaged and reconstructed dendritic arbors to compare dendritic growth rate during a 3 hr dark-exposure period vs. growth rate after 3 hr of enhanced visual stimulation. We observed that in the control group, enhanced visual stimulation results in an increase in the total dendritic branch length compared to the dark period [*Growth of total dendritic length (percent): Control/Dark, 97.35 ± 5.76, n = 9; Control/Light, 128.32 ± 7.25, n = 9*] (*Figure 5C*). In the presence of SB-3CT, the enhanced visual stimuli does not increase total dendritic branch length [*Growth of total dendritic length (%): SB-3CT/Dark, 107.30 ± 6.61, n = 9; SB-3CT/Light, 96.88 ± 7.88, n = 9*]. Thus, the change in dendritic growth rate was significantly decreased in the SB-3CT reared tadpoles in comparison to the matched control animals [*Change in growth rate (%): Control, 136.61 ± 12.29, n = 9; SB-3CT, 91.27 ± 6.42, n = 9; p = 0.009*] (*Figure 5D,E*). Taken together, these results suggest that enhanced visual stimulation increased MMP-9 levels, and that this increase is necessary for proper activity-driven dendritic arborization. By extension, chronically elevated levels of MMP-9 present in VPA-treated tadpoles would result in unregulated plasticity, leading to hyperconnectivity and hyperexcitability.

These findings, indicating that MMP9 is required for structural plasticity, and that MMP9 is required for synaptic changes induced by VPA exposure, suggest that MMP9 may be targeting ECM elements that normally stabilize dendritic architecture. Perineuronal nets (PNNs) are lattice-like structures that cover soma and proximal dendrites. Studies in rodent models of developmental plasticity have shown that MMP9 cleaves various components of perineuronal nets to modify neuronal plasticity (*Murase et al., 2017*). We used fluorescein tagged Wisteria floribunda agglutinin (WFA) to visualize the distribution of chondroitin sulfate proteoglycans (CSPGs) component of PNNs across *Xenopus* tectum for VPA/control-treated animals (*Murase et al., 2017*). *Xenopus* tectum consists of a well-defined layer of cell bodies extending their dendrites to the neuropil layer. We studied the WFA distribution across the cell body and neuropil layers of the *Xenopus* tectum. Tadpoles exposed to VPA had significantly less WFA intensity in the cell body layer in comparison to controls, suggesting potential role of MMP9 in mediating proteolysis of PNNs (*Figure 5—figure supplement 1*). The WFA intensity for the neuropil layer was not significantly different across control/VPA groups, possibly due to its more diffuse structure. This suggests that PNNs are a potential downstream target of MMP9 in this system, opening the door to further lines of inquiry.

## Discussion

The results of this study suggest that chronic MMP-9 upregulation during early exposure to VPA, is responsible for mediating its effects on neural circuit connectivity and behavior. Exposure to VPA during a key developmental period in *Xenopus* produces hyperconnected and hyperexcitable neural networks responsible for producing abnormal behaviors such as enhanced seizure susceptibility and decreased habituation (*James et al., 2015*). Similarly, we found that upregulation of MMP-9 via overexpression resulted in increased frequency of sEPSCs, sIPSCs and barrages suggesting increased tectal connectivity and excitability. Downregulation of MMP-9 not only reversed VPA-induced synaptic connectivity and excitability in developing *Xenopus* tectum but also rescued VPA-induced behavioral effects. This suggests that chronically elevated levels of MMP-9 may produce

abnormally high and sustained plasticity, leading to overconnectivity. Sensory experience during critical periods of development is crucial for proper maturation of sensory networks, and may involve transient, activity-dependent increases in MMP-9. In *Xenopus*, acute exposure to enhanced visual activity results in increased plasticity and dendritic growth in optic tectal neurons (*Aizenman and Cline, 2007*; *Sin et al., 2002*). We found that following enhanced visual stimulation, MMP-9 levels were elevated and that increased MMP-9 was necessary for experience-dependent dendritic growth, suggesting that during normal development, MMP-9 levels are low but transiently elevated to facilitate activity-induced structural plasticity.

MMP-9 plays a paramount role in establishing synaptic connections during development and in restructuring of synaptic networks in the adult brain (*Reinhard et al., 2015*). Recent studies show that MMP-mediated proteolysis is important for remodeling of synaptic structure and function required for processes such as learning and memory, and action of MMPs on various extracellular matrix components, cell surface receptors and growth factors is crucial for modulating neurogenesis and neuroplasticity (*Huntley, 2012*). Dysregulated expression and activity of MMP-9 can result in abnormal synaptic remodeling; a hallmark of various neurodevelopmental disorders (*Huntley, 2012*). In particular, dysregulation of MMP-9 is known to influence cognitive processes (*Beroun et al., 2019*) and increased MMP-9 levels are associated with several neurodevelopmental disorders including ASD (*Abdallah et al., 2012*; *Yoo et al., 2016*), fragile X syndrome (*Dziembowska et al., 2013*; *Sidhu et al., 2014*), epilepsy (*Konopka et al., 2013*), bipolar disorder (*Rybakowski et al., 2013*), and schizophrenia (*Yamamori et al., 2013*). Consistent with these studies, our data showed that elevated MMP-9 activity results in neurodevelopmental deficits in *Xenopus* by creating a hyperplastic state which leads to local overconnectivity.

In addition to MMP9, there is also the possibility that other MMPs may be involved in this process. While MMP-9 showed the largest fold increase in the initial microarray screen after chronic VPA exposure, other MMPs with known effects in neural development and brain function (*Beroun et al., 2019*; *Small and Crawford, 2016*; *Werner et al., 2008*) showed significantly altered levels, including MMP25, MMP7, MMP14 and MMP28, with the second largest increase being in MMP25. Interestingly, MMP25 has been implicated in development of sensory neurons in zebrafish (*Crawford et al., 2014*) and it would be interesting to examine hindbrain and tectal innervation by other sensory modalities after VPA exposure (*Deeg et al., 2009*). Although changes in these other MMPs are yet to be confirmed by qPCR, they raise the intriguing possibility that multiple MMPs may be working in concert to regulate development and their dysregulation can also potentially have an impact in neurodevelopmental disorders.

Early life sensory experience during critical periods of development modifies functional circuits in the brain (*Chklovskii et al., 2004*; *Hensch and Fagiolini, 2005*). Periods of high MMP-9 activity correlate with structural and functional synaptic reorganization during critical period plasticity (CPP) and developmental windows across CNS regions, suggesting a role for MMP-9 in determining the developmental time course of CPP (*Kaliszewska et al., 2012*; *Oliveira-Silva et al., 2007*; *Spolidoro et al., 2012*). The results from our study are consistent with these observations in that MMP-9 levels increase in response to enhanced visual activity during a key period of development. Furthermore, this increase in MMP-9 correlates with visual activity induced plasticity and the inhibition of MMP-9 using a pharmacological inhibitor prevents visual-activity-induced dendritic growth. The secretion and activation of MMP-9 is tightly regulated by a wide variety of factors and once activated, it can act on multiple targets including various components of the ECM, cell adhesion molecules, cell surface receptors, cytokines, growth factors, and other proteases, each of which can have differential effects in creating permissive environments for structural plasticity in the brain (*Van den Steen et al., 2002*; *Vandooren et al., 2013*).

This central role for MMP-9 is further supported by recent studies in adult mice, in which upon light reintroduction, intense MMP-9 activity is observed surrounding excitatory synapses in the binocular visual cortex suggesting that MMP-9 can modulate plasticity even after closure of critical periods (*Murase et al., 2019*; *Murase et al., 2017*). Enzymatic activity of MMP-9 has been known for inducing and maintaining long-term potentiation and causes elongation and thinning of dendritic spines in the hippocampal neurons (*Huntley, 2012*; *Michaluk et al., 2011*). MMP-9 knock-out mice and overexpressing models show significantly reduced synaptic plasticity suggesting that appropriate levels of MMP-9 are needed for structural and functional reorganization of the brain (*Wiera et al., 2013*). Local activation of MMP-9 at the level of individual synapses can induce the

maturation of spine heads (*Szepesi et al., 2014*). MMP-9 can act on various cell surface receptors like integrins, BDNF, ICAMs, NGL, all of which are known to be important for developmental and adult plasticity (*Fiore et al., 2002*; *Lee et al., 2014*; *Michaluk et al., 2009*; *Yamamori et al., 2013*). MMP-9, through activation of cell surface and cell adhesion molecules, is implicated not only in active dendritic spine remodeling and stabilization but also in pre- and postsynaptic receptor dynamics; consolidation of LTP; myelination and synaptic pruning (*Reinhard et al., 2015*). Perineuronal nets (PNNs), which are extracellular matrix components, are often associated with inhibitory interneurons and regulate their development and functions. Recent studies have shown that MMP-9 can rapidly degrade the PNNs surrounding various neurons enabling synaptic and network reorganization during experience dependent plasticity, and could also provide a mechanism by which dysregulation of MMP-9 can lead to abnormal development of neural circuitry (*Murase et al., 2019*; *Murase et al., 2017*; *Wen et al., 2018*). As we show in our data, VPA exposure also leads to degradation of PNNs in the developing tectum, suggesting that this is one possible downstream target in this system. Going forward, understanding the precise downstream targets and functions of MMP-9 will be critical for understanding various cellular and molecular mechanisms underlying neurodevelopmental disorders. Our study highlights the importance of MMP-9 in the etiology of neurodevelopmental disorders and future studies will focus on investigating the downstream molecular targets of MMP-9 and the role of other MMPs.

# Materials and methods

## Key resources table

| Reagent type (species) or resource | Designation | Source or reference | Identifiers | Additional information |
|---|---|---|---|---|
| Strain, strain background (*Xenopus laevis*) | Tadpole | Nasco | WT and Albino strains; LM00456, LM00456(A) | Bred in house |
| Chemical compound, drug | Valproic Acid | Sigma | Cat. No. P4543 | Drug |
| Chemical compound, drug | SB-3CT | Tocris | Cat. No. 6088 | MMP9/2 inhibitor |
| Recombinant DNA reagent | pCMV-SPORT6, MMP-9, GFP | Invitrogen | pCMV-SPORT6 (backbone) | Expressing *Xenopus* MMP9 and GFP, or GFP alone, or MMP9-FLAG |
| Genetic reagent | MMP-9 MO | GeneTools *Bestman and Cline, 2014* | DOI:10.1007/978-1-62703-655-9_11 | Morpholino antisense oligonucleotide- lissamine tagged, Used to drive down expression of MMP9 |
| Genetic reagent | Control MO | GeneTools *Bestman and Cline, 2014* | DOI:10.1007/978-1-62703-655-9_11 | Used as a transfection control, Morpholino antisense oligonucleotide-lissamine tagged |
| Antibody | Anti-MMP9 (Rabbit polyclonal) | EMD Millipore | Ab19016; RRID:AB_91090 | WB (1:2000) |
| Antibody | Anti-Rabbit-HRP (goat polyclonal) | BioRad | #1662408EDU; RRID:AB_11125345 | Secondary Ab for WB |
| Commercial assay or kit | Pierce ECL-western blot substrate | Thermo Fisher | Cat#32209 | For visualizing labeling |
| Recombinant DNA reagent | pCALNL-GFP (plasmid) | *Schohl et al., 2020* , DOI: 10.3389/fncir.2020.00047 | | For single cell labeling |
| Recombinant DNA reagent | pCAG-Cre (plasmid) | *Schohl et al., 2020*, DOI: 10.3389/fncir.2020.00047 | | For single cell labeling |

*Continued on next page*

*Continued*

| Reagent type (species) or resource | Designation | Source or reference | Identifiers | Additional information |
|---|---|---|---|---|
| Recombinant DNA reagent | pBE-PSD-95 GFP | *Sanchez et al., 2006*; DOI: 10.1242/dev.02409 | | For synaptic labeling |
| Recombinant DNA reagent | pCMV-mCherry | *Schwartz et al., 2009*; DOI:10.1016/j.neuron.2011.02.055 | | Cell filling mCherry |
| Chemical compound, drug | F-WFA | Vector Labs | FL-1351–2 | Fluorescein-labeled Wisteria floribunda agglutinin, 1:500 dilution for staining PNNs |

All animal experiments were performed in accordance with and approved by Brown University Institutional Animal Care and Use Committee standards and guidelines (Protocol number 19-05-0016).

## Experimental animals

All experimental comparisons were done with matched controls from the same clutch to account for variability in responses over time and across experimental groups.

## Drug treatments

Tadpoles were raised in Steinberg's rearing media on a 12 hr light/dark cycle at 18–21°C for 7–8 days, until they reached developmental stage 42 (*Nieuwkoop and Faber, 1968*). They were then transferred to either (i) control rearing media, (ii) 1 mM solution of VPA in Steinberg's solution, (iii) 1 mM VPA + 3 uM SB-3CT (a pharmacological inhibitor of MMP-9) or (iv)3 uM SB-3CT alone, depending on the experiment, and raised at temperatures ranging from 18°C to 21°C until they reached developmental stages of either 47–48 (15–16 days post-fertilization). Developmental stages of tadpoles were determined according to *Nieuwkoop and Faber, 1968*. For all the electrophysiological and behavioral experiments, the same concentration of VPA and SB-3CT were used. Previous experiments utilizing 1 mM VPA elicited distinct electrophysiological and behavioral effects (*James et al., 2015*). The rearing medium was renewed every 3 days. Animals of either sex were used because, at these developmental stages, tadpoles of either sex are phenotypically indistinguishable.

## Plasmid and antisense constructs

MMP-9 overexpression construct: The MMP-9 overexpression plasmid contained a backbone of pCMV-SPORT6 plasmid with CMV promoter driving the expression of downstream cloned genes - MMP-9 and a GFP reporter. GFP control construct: The GFP control plasmid consisted of a backbone of pCMV-SPORT6 plasmid with CMV promoter driving the expression of only a GFP reporter. Each working solution contained 4 ug/ul of appropriate overexpression plasmid + 0.3 µL Fast Green.

Reduction of matrix metalloproteinase nine gene expression was accomplished through electroporation of a morpholino antisense oligonucleotide (MO) (*Bestman and Cline, 2014*). Lissamine tagged MOs were designed and ordered from GeneTools. Sequences used for the MMP-9 MO and control MO were 5' AGACTAAAACTCCCACCCTACCCAT 3' and 5' CCTCTTACCTCAGTTACAA TTTATA 3', respectively, and were adopted from the methods of *Faulkner et al., 2014*. The control MO was a scrambled MO not known to align elsewhere. Working solution was prepared by diluting stock MO using endotoxin free molecular grade water and adding 1% Fast Green stock solution to make up 6% of the working solution, with a final concentration of 0.1 mM of MO.

## Tectal cell transfection

To transfect cells in the tectum, stage 43–44 animals were anesthetized in a solution of 0.01% MS-222, plasmids or antisense morpholino constructs were injected into the midbrain ventricle, and voltage pulses were applied across the midbrain using platinum electrodes to electroporate cells lining the tectal ventricle (*Haas et al., 2002*). Tadpoles were then screened for appropriate transfection by

checking for fluorescence under a fluorescence microscope. The electroporation protocol used for MMP9 overexpression resulted in widespread transfection across tectal lobes with average 116.83 ± 2.74 labeled neurons per tectal lobe (n=9).

## Behavioral assays

Tadpoles that were used for seizures and acoustic startle response habituation protocols were not used again for other experiments. Immediately before behavioral experiments, tadpoles were transferred to Steinberg's media and left for 1 hr to recover from acute action of the drug.

## Seizures

Seizures were analyzed as described in *James et al., 2015*. For seizure experiments, when tadpoles reached stage 47, they were transferred into individual wells in a six-well plate (Corning), each filled with 7 ml of 5 mm pentylenetetrazol (PTZ) solution in Steinberg's media. The plate was diffusely illuminated from below and an overhead SCB 2001 color camera (Samsung) imaged tadpoles at 30 frames/s. Tadpole positions were acquired using Noldus EthoVision XT (Noldus Information Technology) and processed offline in a custom MATLAB program (MathWorks; *Khakhalin, 2019*; https://github.com/khakhalin/Xenopus-Behavior). Seizure events were defined as periods of rapid and irregular movement, interrupted by periods of immobility (*Bell et al., 2011*). The onset of regular seizures happened on average 3.9 ± 1.3 min into the recording and were detected automatically using swimming speed thresholding at a level of half of the maximal swimming speed. Number, frequency, and duration of seizure events were measured across 5 min intervals of a 20-min-long recording.

## Startle habituation response

A six-well plate fixed between two audio speakers (SPA2210/27; Philips), and mechanically connected to diaphragms of these speakers by inflexible plastic struts acted as an experimental arena for the startle habituation response experiment. Stage 49 tadpoles were placed in the experimental arena and acoustic stimuli (one period of a 200 Hz sine wave, 5 ms long) were presented every 5 s which evoked a reliable startle response. The observed startle response is likely due to a combination of inner ear-mediated and lateral line-mediated inputs, and, for the purposes of this study, we did not attempt to differentiate between the two. The amplitude of stimuli for habituation experiments was set at two times above the startle threshold. Each train of stimuli lasted for 2 min; trains 1 to 5 were separated by 5 min gaps, whereas train six was separated from train 5 by a 15 min gap. In order to compensate for possible variation in stimulus delivery across the wells, the location of treatment (1 mM VPA and 1 mM VPA + 3 uM SB-3CT) tadpoles and matched controls within the six-well plate was alternated across experiments. An overhead camera was used to acquire videos of the tadpoles, which were then tracked in EthoVision and processed offline in a custom MATLAB script (*Khakhalin, 2019*; https://github.com/khakhalin/Xenopus-Behavior). Peak speed of each startle response was measured across a 2 s interval after stimulus delivery. To quantify habituation of startle responses at different timescales, we adopted the nomenclature from previous studies (*Eddins et al., 2010*; *Roberts et al., 2011*). Startle speeds were averaged across 1-min-long periods of acoustic stimulation and compared across periods. For rapid habituation, we compared responses during minutes 1 and 2 of the first 2-min train of stimuli.

## Electrophysiology experiments

For whole-brain recordings, tadpole brains were prepared as described by *Wu et al., 1996* and *Aizenman et al., 2003*. In brief, tadpoles were anesthetized in 0.02% tricainemethane sulfonate (MS-222). To access the ventral surface of the tectum, brains were filleted along the dorsal midline and dissected in HEPES-buffered extracellular saline [in mm: 115 NaCl, 2 KCl, 3 Cacl2, 3 MgCl2, 5 HEPES, 10 glucose, pH 7.2 (osmolarity, 255 mOsm)]. Brains were then pinned to a submerged block of Sylgard in a recording chamber and maintained at room temperature (24°C). To access tectal cells, the ventricular membrane surrounding the tectum was carefully removed using a broken glass pipette. For evoked synaptic response experiments, a bipolar stimulating electrode (FHC) was placed on the optic chiasm to activate RGC axons.

Whole-cell voltage-clamp and current-clamp recordings were performed using glass micropipettes (8–12 MΩ) filled with K-gluconate intracellular saline [in mm: 100 K-gluconate, 8 KCl, 5 NaCl,

1.5 MgCl2, 20 HEPES, 10 EGTA, 2 ATP, and 0.3 GTP, pH 7.2 (osmolarity, 255 mOsm)]. Recordings were restricted consistently to retinorecipient neurons in the middle one-third of the tectum, thus avoiding any developmental variability existing along the rostrocaudal axis (*Hamodi and Pratt, 2014*; *Khakhalin and Aizenman, 2012*; *Wu et al., 1996*). Electrical signals were measured with a Multiclamp 700B amplifier (Molecular Devices), digitized at 10 kHz using a Digidata 1440 analog-to-digital board, and acquired using pClamp 10 software. Leak subtraction was done in real time using the acquisition software. Membrane potential in the figures was not adjusted to compensate for a predicted 12 mV liquid junction potential. Data was analyzed using AxographX software. Spontaneous synaptic events were collected and quantified using a variable amplitude template (*Clements and Bekkers, 1997*). Spontaneous EPSCs (sEPSCs) were recorded at −45 mV (the reversal for chloride ions) whereas sIPSCs were collected in control media at 5 mV (the reversal for glutamatergic currents). For each cell, 60 s of spontaneous activity was recorded. For evoked synaptic response experiments, a bipolar stimulating electrode (FHC) was placed on the optic chiasm to activate RGC axons. Synaptic stimulation experiments were conducted by collecting EPSCs evoked by stimulating the optic chiasm at a stimulus intensity that consistently evoked maximal amplitude EPSCs. Polysynaptic stimulation experiments were performed by collecting EPSCs evoked by stimulating the optic chiasm at a stimulus intensity that evoked the maximal amplitude EPSC. Quantification of polysynaptic activity was calculated by measuring the total change in current over 100 ms time bins beginning at the onset of the evoked response. A spontaneous barrage was defined as a change in holding current of 10 or 20 pA intervals for a period of >200 ms (*James et al., 2015*). The sEPSC recordings were scanned carefully manually to detect and quantify the number of barrages observed for each of the experimental groups. Graphs show median and interquartile ranges (IQRs) as error bars, and data in the text shows the averages and SEs. Across experiments we did not find any significant differences in either input resistance or cell capacitance between groups, as indicated in *Supplementary file 1*.

## Real-time quantitative PCR analysis

For quantifying MMP9 expression levels, whole tecta were harvested at developmental stage 49 from tadpoles reared in control conditions or with 1 mM VPA from stage 42. Sequences used for the MMP-9 forward and reverse primer were 5' TCATGGGATCTTTTGCTCGT 3' and 5' CCTTGGG TAGCCATTATCAA 3', respectively. Optic tecta were collected from 10 tadpoles from three independent breedings and stored in RNAlater prior to the isolation of RNA by TRIzol extraction. Super-Script IV VILO first strand cDNA synthesis was then performed according to the manufacturer's protocol (Invitrogen). Real-time quantitative PCR (RT-qPCR) was performed by using Power SYBR Green master kit with an Applied Biosystems StepOnePlus according to standard manufacturer's protocols (ThermoFisher). Data were analyzed by the ΔΔCT relative quantification method using the housekeeper gene RSP13.

## Quantification of MMP-9 protein levels

Western blots were used to examine the protein expression changes in the midbrain of the tadpoles under different visual experience conditions or the knockdown of MMP-9. To examine the specificity of MMP-9 morpholino, whole-brain electroporation was performed at stage 47 tadpoles with 0.4 mM morpholino oligos and plasmid that expresses MMP-9 fused with flag peptide (1 mg/ml), and the midbrain tissues were dissected 2 days later. To test the effect of morpholino on knocking down endogenous MMP-9 protein expression, whole-brain electroporation was performed at stage 47 tadpoles with 0.4 mM morpholino oligos, and the mid-brain tissues were collected 2 days after electroporation. In order to test the effect of different visual experience conditions on MMP-9 protein expression, animals were reared under enhanced visual stimulation, dark or ambient light (12 hr enhanced visual stimulations/12 hr dark).

Tissues were homogenized in lysis buffer documented in *Joukov et al., 2001* (HEPES 100 mM (pH 7.5), NaCl 200 mM, EDTA 40 mM, EGTA 4 mM, NaF 100 mM, β-glycerophosphate 20 mM, sodium orthovanadate 2 mM, Nonidet P-40 1%, Complete Protease Inhibitor mixture 1:50), and the protein concentration was measured with Bio-Rad DC Protein Assay kit. Twenty μg of lysate was loaded onto an in-house made 7% gel for electrophoresis. Proteins were transferred to a nitrocellulose membrane with Trans-Blot Turbo transfer system (BioRad). The membrane was incubated in 5%

blotting reagent/0.1% Tween-20 (Sigma) in TBS for an hour for blocking, and then transferred to primary MMP-9 antibody solution (1:2 k; EMO millipore ab19016) diluted in blocking solution and incubated overnight at 4°C. After three brief washes with 0.1% Tween-20 in TBS, membranes were transferred to goat anti-rabbit HRP-conjugated secondary antibody (BioRad), diluted in blocking solution for an hour at room temperature. The Pierce ECL western blot substrate (Thermo Fisher Scientific, 32209) was used to visualize labeling. For quantification analysis, different exposure periods were used for the same blots to avoid saturation, and total protein normalization was made to Ponceau S staining as a loading control (*Romero-Calvo et al., 2010*). The densitometric analysis of western blot and the total protein was performed using Gel Analyzer of ImageJ, and the data of MMP-9 protein expression were normalized against the total protein as a loading control. Staining using anti-MMP-9 antibodies often resulted in labeling some non-specific bands in addition to the 75 kDa band. Thus, MMP-9-flag overexpression was used in combination with anti-flag antibodies to confirm specificity of the MMP-9 MO and confirm location of MMP-9 band.

## Neuronal morphology

To label single tectal cells, Stage 44–45 tadpoles were electroporated using the pCALNL–GFP plus pCAG–Cre plasmids (courtesy Ed Ruthazer, McGill University, Montreal, Quebec, Canada) (*Ruthazer et al., 2013*) at a ratio of 5000:1 to limit coexpression to a very small subset of neurons. Approximately 3 days after electroporation (around developmental stages 48–49), the animals were screened to check for successful fluorescence. Tadpoles were selected for those in which a single neuron in the tectum was expressing GFP and could be imaged clearly. A Zeiss LSM 800 Confocal Laser Scanning Microscope was used to image tadpoles containing labeled neurons. Prior to the experiment, tadpoles were either immersed in control media or media containing SB3CT. Animals were then anesthetized in 0.02% MS-222 and imaged under the 40X water immersion objective. For each animal, the clearest, most isolated and identifiable cell in the tectum was imaged. Initially, a z-stack encompassing the cell body and the entire length of all neurites was taken for each animal (as the reference image). After these reference images were taken, the tadpoles were transferred to the rearing media (Control or 3 uM SB3CT) and incubated in dark for 3 hr. The same cell that had been previously imaged was identified for each animal and imaged after the 3 hr dark exposure. The tadpoles were then put back into their respective wells and subjected to enhanced visual stimuli using a 'light box' for 3 hr. The light box consisted of a chamber with 4x3 array of green LEDs attached to the ceiling. Each row of LEDs flashed for 1 s and was then followed by the adjacent row, until all four rows had flashed and a 1 s pause occurred. The cycle then started over, and was presented over a 3 hr period. After visual simulation, a third image was taken. Dendritic arbors of imaged neurons were reconstructed in 3D and analyzed using Neutube and Fiji software (respectively). The total path length of each image was measured. Then, using the series of three images taken for each cell, growth rate was calculated.

## In vivo expression of PSD95-GFP and synapse density analysis

We used a pBE-PSD95-GFP containing plasmid in combination with pcDNA-mCherry (courtesy Ed Ruthazer, McGill University, Montreal, Quebec, Canada) in *Xenopus* tectum to visualize dendritic processes to allow us to quantify synapse density in control and VPA groups. Stage 42 tadpoles were exposed to VPA/Steinberg's solution until they reached stage 47–48. At stage 44–45, tadpoles were electroporated with PSD95-GFP and pcDNA-mCherry (1:1 proportion) using standard electroporation protocol. Once tadpoles reached stage 49 (approximately 4–5 days post electroporation), they were screened for well isolated tectal cells/regions with mCherry and punctate GFP expression. Tadpoles were anesthetized and isolated neurons/regions with well-defined puncta were imaged with a 40x objective (40x Water, NA 1.2) in a Zeiss LSM 800 Confocal Laser Scanning Microscope (with 4x averaging). For each tadpole, optical sections (0.5 µm thick) of well isolated neurons/regions were gathered simultaneously at two wavelengths (green and red) to get precise identity and position of synaptic processes along the dendritic arbor. PSD95-GFP punctate distribution was used to quantify post-synaptic processes, while pcDNA-mCherry expression was used to trace neuronal morphology. Data analysis for quantification of GFP-puncta was performed on raw z-stack images using the 3D Objects Counter plug-in Fiji. All the images had distinct punctate GFP expression along dendritic arbors with some background non-punctate GFP. An in-built thresholding function of 3D

Objects Counter was used to detect discrete individual puncta. The criteria used for thresholding involved subtracting pixel values of background non-punctate GFP along the dendritic arbors. Any background non-punctate GFP in the cell body and surrounding region was excluded from the data analysis. Various parameters like total number of puncta, size, intensity, etc. were obtained from the 3D Objects Counter plugin output. For the same image, dendritic arbors were reconstructed in 3D and analyzed using the Simple Neurite Tracer plugin in Fiji to measure various parameters including total dendritic length. The ratio of total number of puncta (obtained from 3D Objects Counter) and total dendritic length (obtained from Simple Neurite Tracer) was used to calculate synapse density.

### Detection and quantification of perineuronal nets (PNNs)

Tadpoles were exposed to VPA/Steinberg's solution at stage 42 until they reached stage 47–48. A fluorescein-tagged Wisteria floribunda agglutinin (F-WFA) was used to visualize the distribution of the chondroitin sulfate proteoglycan (CSPGs) component of PNNs across *Xenopus* tectum for VPA/control treated animals. At stage 49, tadpoles were anesthetized, fixed in 4% paraformaldehyde for 24 hr followed by serial sucrose gradient (overnight incubation with 15%, 20%, and 30% sucrose solution each). The tadpoles were then frozen in OCT and thin (20 µm) horizontal sections were obtained using cryostat (Leica). Sections were blocked in 3% normal donkey serum (NDS) in 1X PBS for 1 hr at room temperature. Sections were then incubated in 1:500 dilution of F-WFA (Vector Labs FL-1351–2, Fluorescein labeled Wisteria Floribunda Lectin, WFA, WFL) in 1X PBS with 2% NDS at 4° C for 20–24 hr. F-WFA incubation was followed by nuclei staining using DAPI and slides were mounted with DPX for imaging. Sections were imaged with a 40x (1.2 NA) objective in a Zeiss LSM 800 Confocal Laser Scanning Microscope (with 4x averaging) and the distribution of PNNs in the optic tectum was analyzed. *Xenopus* tectum consists of a well-defined layer of cell bodies extending their dendrites to the neuropil layer. F-WFA staining showed diffused distribution of CSPGs across the cell body and neuropil layer. Occasionally, we observed intact PNNs surrounding specific cell bodies. Since the majority of images contained diffuse staining, we decided on using average F-WFA intensity as a measure for PNNs distribution. Maximum intensity projections were obtained for each image and average intensity values were measured for cell body and neuropil region using Fiji. Average F-WFA intensity for a tadpole was calculated by averaging the F-WFA intensity of at least three sections from that tadpole. In order to minimize error in F-WFA intensity for every slide, we separated the slide in two halves, and placed sections from each control and VPA in each half. Introduction of Chondroitinase ABC (Sigma, 9024-13-9) results in degradation of PNNs, which was seen by decrease in F-WFA intensity in tectum and was used as a positive control in all the experiments.

### Statistical analyses

For all the data, averages and SEs are given in the text, whereas median values and interquartile range (IQR) are shown in the figures. Sample sizes were based on power analyses and known variability from prior work in our experimental system. No outliers were removed from the data analysis; however, data points that were above 1.5 times IQR may be indicated as outliers in some graphs for data presentation purposes and increased clarity. One-way ANOVA was used to compare differences between overall values of the groups. Repeated measures ANOVA was used to compare differences between startle speed (cm/s) with treatment groups as between-subjects variables and bouts as a repeated measure to assess habituation response upon repeated stimulation. Student two-tailed t-test with unequal variances was used to assess differences in relative MMP-9 levels across different treatments. Differences were considered statistically significant at $p < 0.05$, and significant results followed by Bonferroni post-hoc pairwise comparisons. All statistical analyses were carried out in R (R v3.6.1, http://www.R-project.org) with supplementary installed packages 'readr','lsr', 'ggplot2', 'nlme', 'ez','dplyr', 'stats', and 'lme4', 'emmeans'.

## Acknowledgements

We are grateful for valuable intellectual input from members of the Aizenman lab. We thank Phouangmaly Mimi Oupravanh and Virgilio Lopez for animal and lab care. This work was supported by a Brown University OVPR Seed Award, a Carney Institute RI Neuroscience New Frontiers Award,

NSF GRFP (EJ), and NIH – NEI R01 EY027380. We thank Dr. Ed Ruthazer (McGill University, Montreal, Quebec, Canada) for providing plasmid constructs for labeling PSD95-GFP.

## Additional information

### Funding

| Funder | Grant reference number | Author |
|---|---|---|
| National Science Foundation | GRFP | Eric J James |
| National Eye Institute | R01 EY027380 | Carlos Aizenman<br>Sayali V Gore<br>Adrian Thompson |
| Brown University | Carney New Frontiers and OVPR SEED award | Carlos Aizenman |
| National Eye Institute | R01 EY011261 | Hollis T Cline<br>Lin-chien Huang |
| National Institute of Neurological Disorders and Stroke | NS076006 | Hollis T Cline<br>Lin-chien Huang |

The funders had no role in study design, data collection and interpretation, or the decision to submit the work for publication.

### Author contributions

Sayali V Gore, Conceptualization, Data curation, Formal analysis, Validation, Investigation, Methodology, Writing - original draft, Writing - review and editing, designed experiments, performed electrophysiology experiments, analyzed data, prepared figures, ran statistics and prepared the manuscript; Eric J James, Conceptualization, Formal analysis, Investigation, performed electrophysiology experiments; Lin-chien Huang, Conceptualization, Resources, Formal analysis, Validation, Investigation, Methodology, Writing - review and editing, designed and performed western blot analysis for quantifying MMP-9 levels and designed morpholinos, and contributed to the writing; Jenn J Park, Conceptualization, Formal analysis, Investigation, performed the behavioral assays; Andrea Berghella, Formal analysis, Investigation, Methodology, performed neuronal morphology experiments; Adrian C Thompson, Formal analysis, Validation, Investigation, Methodology; Hollis T Cline, Conceptualization, Supervision, Funding acquisition, Writing - review and editing, designed experiments for western blot analysis for quantifying MMP-9 levels and designed morpholinos, and contributed to the writing and project design/conceptualization; Carlos D Aizenman, Conceptualization, Formal analysis, Supervision, Funding acquisition, Investigation, Visualization, Writing - original draft, Project administration, Writing - review and editing, oversaw project and design, performed electrophysiology experiments, prepared manuscript

### Author ORCIDs

Hollis T Cline https://orcid.org/0000-0002-4887-9603
Carlos D Aizenman https://orcid.org/0000-0002-7378-7217

### Ethics

Animal experimentation: All animal experiments were performed in accordance with and approved by Brown University Institutional Animal Care and Use Committee standards and guidelines (Protocol number 19-05-0016).

### Decision letter and Author response

Decision letter https://doi.org/10.7554/eLife.62147.sa1
Author response https://doi.org/10.7554/eLife.62147.sa2

## Additional files

### Supplementary files

- Supplementary file 1. Intrinsic cellular properties in different experimental groups. Input resistance and membrane capacitance values showed no significant differences between experimental groups.

- Transparent reporting form

### Data availability

All data generated or analysed during this study are included in the manuscript and supporting files.

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
