## [Decision Letter]

**Acceptance summary:**

Previous work from the Aizenman lab demonstrated the utility of the *Xenopus* tectum as a model to examine neuronal, circuit and behavioral manifestations of valproic acid (VPA) treatment, a teratogen associated with autism spectrum disorder in humans. In the current manuscript they demonstrate that the deficits induced by VPA treatment, including enhanced spontaneous and evoked neuronal activity, are blocked by pharmacological or morpholino based inhibition of MMP9. Inhibition of MMP9 also reverses the effects of VPA treatment on seizure susceptibility and the startle habituation response. Over-expression of MMP9 pheno-copies the effect of VPA, and inhibition of MMP9 in single tectal neuronal blocks the expression of experience-dependent structural plasticity. Overall, this is a convincing demonstration of the impact of MMP9 in the regulation of experience dependent structural plasticity in a model system. Of note, the authors are advised to state the potential caveats of the PSD-95 expression approach as pointed about in the re-review comments.

**Decision letter after peer review:**

Thank you for submitting your article "Role of matrix metalloproteinase-9 in neurodevelopmental disorders and plasticity in *Xenopus* tadpoles" for consideration by *eLife*. We apologize that it took longer than we hope for the review as it took longer than expected to secure reviewers. Your article has now been reviewed by 3 peer reviewers, one of whom is a member of our Board of Reviewing Editors, and the evaluation has been overseen by Gary Westbrook as the Senior Editor. The reviewers have opted to remain anonymous. The reviewers have discussed the reviews with one another and the Reviewing Editor has drafted this decision to help you prepare a revised submission.

The editors and reviewers have judged that your manuscript is of interest, but as described below that additional experiments are required before it is published. We would like to draw your attention to changes in our revision policy that we have made in response to COVID-19 (https://elifesciences.org/articles/57162). First, because many researchers have temporarily lost access to the labs, we will give authors as much time as they need to submit revised manuscripts. We are also offering, if you choose, to post the manuscript to bioRxiv (if it is not already there) along with this decision letter and a formal designation that the manuscript is "in revision at *eLife*". Please let us know if you would like to pursue this option. (If your work is more suitable for medRxiv, you will need to post the preprint yourself, as the mechanisms for us to do so are still in development.)

Summary

The Aizenman lab has previous demonstrated the utility of the *Xenopus* tectum as a model to examine neuronal, circuit and behavioral manifestations of valproic acid (VPA) treatment, a teratogen associated with autism spectrum disorder in humans. In Gore et al., they demonstrate that the deficits induced by VPA treatment, including enhanced spontaneous and evoked neuronal activity, are blocked by pharmacological or morpholino based inhibition of MMP9. Inhibition of MMP9 also reverses the effects of VPA treatment on seizure susceptibility and the startle habituation response. Over-expression of MMP9 pheno-copies the effect of VPA, and inhibition of MMP9 in single tectal neuronal blocks the expression of experience-dependent structural plasticity. Although the reviewers expressed enthusiasm for the manuscript there were concerns that need to be addressed.

Essential revisions:

1. What is the exact nature of "increased connectivity"? Is there an increase in synapse numbers or solely an increase in dendritic complexity coupled with a functional plasticity? The authors should document properties of mEPSCs and mIPSCs recording in TTX to isolate synaptic properties. Coupling this "mini" analysis to quantification of synapse numbers will address whether the changes are solely due to structural plasticity or also due to a functional potentiation of transmission. These experiments should at least be conducted in MMP-9 overexpression, VPA treatment and VPA treatment+MMP-9 loss-of-function cases to validate the basic premise that there is an increased connectivity.

2. How is increased MMP-9 produces the synaptic and behavioral effects? What is the downstream target (specific receptor?) that would produce the broad changes in synaptic and behavioral phenotypes? Or is this a rather non-specific effect of extracellular matrix? Based on years of data on MMP-9 function its impact on "structural plasticity" in general terms is not surprising but further mechanistic details and specific targets would help move this field forward.

3. The authors refer to microarray data as the rationale for pursuing the role for MMP9 in VPA-induced hyperconnectivity. Please cite the microarray study(ies?). How many other MMPs or proteases with documented roles in development are similarly upregulated? The authors should say how other possible candidate genes did or did not change, perhaps presenting the list with data in a table (at least other MMPs and proteases). If others have changed, the authors should discuss their data in that context.

4. The authors should comment on the specificity of the SB-3CT, particularly with regard to other MMPs or proteases that may/may not have been found to be upregulated in the microarray experiment. Importantly, does SB-3CT rescue the expression levels of MMP-9?

5. The finding that a small number of MMP9 overexpressing cells is fascinating. Have the authors stained the tissue for MMP9 after VPA?

6. Results, first paragraph: although it is in the methods, please state briefly the timing of the VPA exposure and MMP9 overexpression and the age/stage at which the experiments were performed. Within the methods, please give approximate age in days after hatching for the non-tadpole experts.

7. Do the authors have data on the intrinsic cell properties (input resistance, capacitance, etc.)? If so, they should include that data either in Supplemental information or in the text. These factors could absolutely influence hyperconnectivity or measurements of the synaptic properties, so at least the authors should discuss their findings in the context of the findings of James, et al.

8. Please report the number (or %) of tectal neurons in which MMP9 was over-expressed following whole-brain electroporation (Figure 1). -Does MMP9 transfection change the E/I ratio, as previously reported for VPA? Does VPA or MMP9 inhibition change the initial large amplitude/short latency evoked response?

9. please report statistics for total number of barrages or barrage distribution across experimental groups (latter also for Figure 3).

10. Figures3 and 5: The presentation of the immunoblots should clarify if raw or normalized (to Ponceau Blue) data were quantified.

11. Figure 4: Please report a post hoc comparison following the repeated measures ANOVA

12. Figure 5: Total growth and growth rates could also be included in the Results section.

---

## [Author Response]

Essential revisions:1. What is the exact nature of "increased connectivity"? Is there an increase in synapse numbers or solely an increase in dendritic complexity coupled with a functional plasticity? The authors should document properties of mEPSCs and mIPSCs recording in TTX to isolate synaptic properties. Coupling this "mini" analysis to quantification of synapse numbers will address whether the changes are solely due to structural plasticity or also due to a functional potentiation of transmission. These experiments should at least be conducted in MMP-9 overexpression, VPA treatment and VPA treatment+MMP-9 loss-of-function cases to validate the basic premise that there is an increased connectivity.

The reviewer raises an interesting set of questions, and the parameter space here is very large. In fact this experimental question is the topic of a separate set of studies we are currently working on, which involve a combination of serial EM reconstructions, neuronal morphology and electrophysiology.

Here, we have used some of this work to address the reviewer’s questions in combination with prior published data. In terms of the mini analysis proposed by the reviewer it is not clear whether this experiment will yield enough additional information, in the past our analysis of mEPSCa has closely matched our analysis of sEPSCs. We imagine the reason for proposing this was to distinguish a change in frequency caused by elevated network activity vs one caused by having a proliferation of synapses, however both things may be occurring simultaneously, so a change observed in mini frequency would not necessarily rule out that any changes in network activity are not occurring. Furthermore, a change in connectivity is not necessarily expected to result in a change in dendritic complexity. In James et al. 2015, we had shown that VPA exposure alters branching patterns of tectal dendrites while not necessarily changing complexity.

Thus, we used a different metric of synapse number by overexpressing fluorescently tagged postsynaptic marker PSD-95 in control and VPA treated animals. This experiment was a lot more time consuming than we anticipated, taking several months, and we found that VPA exposure does increase the density of PSD-95 labelled puncta. This, combined with the sEPSC observation, indicates that VPA exposure results in increased synapse density, and that sEPSCs are a good proxy for this and can be used for the MMP9 specific experiments in this paper. A more comprehensive examination of changes in synaptic connectivity, presynaptic function, dendritic structure and the role of MMP9 in this process is beyond the scope of this paper, and subject of another set of major studies in the lab. We have incorporated the PSD-95 findings into the text and added them as a supplementary figure:

A PSD95-GFP containing-plasmid was overexpressed in combination with pcDNA-mCherry in *Xenopus* tectum to visualize dendritic processes to allow us to quantify synapse density in control and VPA groups (Sanchez et al., 2006). PSD95-GFP construct has been used previously in *Xenopus* tectum and shows a punctate distribution which is very similar to the endogenous presynaptic SNAP-25 punctate pattern, indicating its location at synaptic sites. Exposure to VPA/control was initiated at stage 42; when tadpoles reached stage 44-45, they were electroporated with PSD95-GFP and pcDNAmCherry. Tadpoles were screened at stage 49 for well-isolated tectal neurons coexpressing mCherry and punctate GFP distribution. We quantified GFP puncta using FIJI plugin (3D object counter) and simultaneously measured dendritic length (using simple neurite tracer) to calculate synapse density (#GFP puncta divided by total dendritic length in microns).

Our results indicate that VPA-exposed tadpoles show increased tectal synapse density compared to control. These results in combination with the electrophysiological experiments suggest that the increased connectivity arises due to increased density of post-synaptic processes.

This finding is mentioned in the Results section, prior to the behavioral results (p. 13).

2. How is increased MMP-9 produces the synaptic and behavioral effects? What is the downstream target (specific receptor?) that would produce the broad changes in synaptic and behavioral phenotypes? Or is this a rather non-specific effect of extracellular matrix? Based on years of data on MMP-9 function its impact on "structural plasticity" in general terms is not surprising but further mechanistic details and specific targets would help move this field forward.

Like the question before, this is a very open ended question with a large parameter space and currently the object of two separate ongoing large projects in the lab. Likely MMP9 is acting on multiple targets to coordinate its effect on structural plasticity, including ECM proteins, growth factor activation and trans-synaptic signalling complexes. To begin to address this within the context of this paper and in response to the reviewer’s comments, we have added data to the effect that VPA exposure results in breakdown of peri-neuronal nets, something which would promote circuit remodelling. Since these findings are not part of the main points being made in the paper, they have been added as a supplementary, and mentioned at the end of the Results section as a potential future direction. A more complete study of PNNs, both during development, plasticity and VPA exposure and the role of MMP9 activation in this process will constitute a future paper in its own right:

Described in Results (p. 24) and discussion (p. 29): MMP9 is a versatile molecule known to regulate pericellular environment by local proteolytic action on various ECM components. We are currently working on identifying downstream molecular targets of MMP9. Perineuronal nets (PNNs) are lattice-like structures that cover soma and proximal dendrites. Studies in rodent models of developmental plasticity have shown that MMP9 cleaves various components of perineuronal nets to modify neuronal plasticity (Murase et al., 2017). We used fluorescein tagged Wisteria floribunda agglutinin (F-WFA) to visualize the distribution of chondroitin sulfate proteoglycans (CSPGs) component of PNNs across *Xenopus* tectum for VPA/control treated animals. *Xenopus* tectum consists of a well defined layer of cell bodies extending their dendrites to the neuropil layer. We studied the WFA distribution across the cell body and neuropil layers of the *Xenopus* tectum. Tadpoles exposed to VPA had significantly less WFA intensity in the cell body layer in comparison to controls; suggesting potential role of MMP9 in mediating proteolysis of PNNs. The WFA intensity for the neuropil layer was not significantly different across control/VPA groups, possibly due to its more diffuse structure.

3. The authors refer to microarray data as the rationale for pursuing the role for MMP9 in VPA-induced hyperconnectivity. Please cite the microarray study(ies?). How many other MMPs or proteases with documented roles in development are similarly upregulated? The authors should say how other possible candidate genes did or did not change, perhaps presenting the list with data in a table (at least other MMPs and proteases). If others have changed, the authors should discuss their data in that context.

We did identify changes in other MMPs that were not as high as in MMP9 and we have added this to the discussion. It is important to know that these other MMPs have not yet been validated by qPCR like we did with MMP9:

“In addition to MMP9, there is also the possibility that other MMPs may be involved in this process. […] Although changes in these other MMPs are yet to be confirmed by qPCR, they raise the intriguing possibility that multiple MMPs may be working in concert to regulate development and their dysregulation can also potentially have an impact in neurodevelopmental disorders.”

We have included short description about the role of various MMPs in Discussion section (p. 27).

4. The authors should comment on the specificity of the SB-3CT, particularly with regard to other MMPs or proteases that may/may not have been found to be upregulated in the microarray experiment. Importantly, does SB-3CT rescue the expression levels of MMP-9?

SB-3CT is a specific inhibitor of MMP9 and MMP2 (Brown et al. 2000). In our microarray screen we did not find alterations in MMP2 levels, and baseline application of SB-3CT had no effect on synaptic transmission (Figure 2), thus it is unlikely that the effect of SB-3CT in reversing VPA-induced changes is due to inhibition of MMP2. Furthermore, the observations that the morpholino experiment replicates the effects of SB-3CT and that antisense MO was made specifically for MMP9, further indicate that MMP2 is not involved. While SB-3CT would not be expected to rescue levels of MMP9 expression (since it is an inhibitor) application of the MO does result in a measurable difference in MMP9 protein levels, as shown in Figure 3A. Combined, the inhibitor data plus the MO data provide strong evidence that a specific decrease in MMP9 activity can reverse the effects of VPA exposure.

5. The finding that a small number of MMP9 overexpressing cells is fascinating. Have the authors stained the tissue for MMP9 after VPA?

It is unclear what the reviewer is referring to in this case, we are assuming that they are referring to the fact that a small number of overexpressing cells can result in network effects in Figure 1.On one hand we found this surprising as well as we did not expect to see network-wide effects. On the other hand, this was not completely unexpected, as we were recording from transfected neurons, and since MMP9 is secreted, it would be expected to have an effect on its local network of surrounding cells. In terms of the second question, we have not done this, but we have qPCR data indicating elevated MMP9 expression levels in the tectum after VPA exposure, but no finer scale data beyond that.

6. Results, first paragraph: although it is in the methods, please state briefly the timing of the VPA exposure and MMP9 overexpression and the age/stage at which the experiments were performed. Within the methods, please give approximate age in days after hatching for the non-tadpole experts.

We have mentioned the timing for VPA exposure/MMP9 overexpression in the first paragraph of results (p. 6). We have also included the approximate age in days in our methods section (p. 32).

7. Do the authors have data on the intrinsic cell properties (input resistance, capacitance, etc.)? If so, they should include that data either in Supplemental information or in the text. These factors could absolutely influence hyperconnectivity or measurements of the synaptic properties, so at least the authors should discuss their findings in the context of the findings of James, et al.

We did not observe any significant difference in input resistance and capacitance between groups, these data have been included in a supplementary table. This is consistent with James et al., where these changes were not observed. Furthermore in James et al. there was a slight decrease in intrinsic excitability (as measured by spike output) in the VPA treated group, suggesting that the changes in spontaneous activity are not driven by increased cellular excitability. This has been noted in the methods (p. 37) and added as a Supplementary Table 1.

8. Please report the number (or %) of tectal neurons in which MMP9 was over-expressed following whole-brain electroporation (Figure 1). -Does MMP9 transfection change the E/I ratio, as previously reported for VPA? Does VPA or MMP9 inhibition change the initial large amplitude/short latency evoked response?

Although expression levels were not quantified for every tadpole, from initial measurements the electroporation protocol used for MMP9 overexpression resulted in widespread transfection across tectal lobes with average 116.83 ± 2.74 (n=9) labelled neurons per tectal lobe (p. 33).

In response to the reviewer’s second question, we examined changes in excitation and inhibition (E:I ratio) by quantifying evoked responses by stimulating optic chiasm using a bipolar electrode. Consistent with previously reported effects of VPA (James et al. 2015), we did not observe any significant difference in E:I ratio across MMP9 overexpression and GFP control groups (E:I ratio: MMP9 overexpression n = 9, 1.14 ± 2.12, GFP control n = 9, 0.31 ± 1.70, p = 0.393). In terms of measuring the early latency response, we also looked at values for total charge over the first 50ms window from evoked responses of excitation and was not significantly different for GFP control and MMP9 overexpression (MMP9 overexpression n = 9, -1146 ± 1030, GFP control n = 9, -577.22 ± 409, p=0.143).

9. please report statistics for total number of barrages or barrage distribution across experimental groups (latter also for Figure 3).

The mean± standard error for the number of barrages is reported throughout the text where appropriate. We did not perform any statistical comparisons on the barrage data. Since these are discrete (vs. continuous) data points, the number of cells is not sufficiently high for a Fisher’s exact test. However we felt it was of value to still report the numbers of barrages observed as distributions in the relevant figures, as this could be informative to the reader and indicative of enhanced network activation. However, the increased spontaneous activity shown throughout, and evoked polysynaptic activity in figures 2 and 3 also are consistent with network hyperexcitability. If the reviewer feels strongly about this, we can remove the barrage data as it is not essential to our conclusions which are already supported by the sEPSC and polysynaptic activity.

10. Figures 3 and 5: The presentation of the immunoblots should clarify if raw or normalized (to Ponceau Blue) data were quantified.

Immunoblots in figures 3 and 5 were normalized to the Ponceau Blue staining, this is indicated in the methods (p. 38).

11. Figure 4: Please report a post hoc comparison following the repeated measures ANOVA

Dunnett’s post-hoc comparison indicates significant differences in SB3CT + VPA vs VPA starting with bout 2.This has been indicated in text (p. 18) and figure (p. 20).

12. Figure 5: Total growth and growth rates could also be included in the Results section.

These are included in the Results section on page 21.